# An Approach for Diver Passive Detection Based on the Established Model of Breathing Sound Emission

Qiang Tu [1], Fei Yuan [1] , Weidi Yang [2] and En Cheng [1,*]

[1]  Key Laboratory of Underwater Acoustic Communication and Marine Information Technology, Ministry of Education, Xiamen University, Xiamen 361005, China; tuqiang@stu.xmu.edu.cn (Q.T.); yuanfei@xmu.edu.cn (F.Y.)

[2]  College of Ocean and Earth Sciences, Xiamen University, Xiamen 361102, China; wdyang@xmu.edu.cn

*  Correspondence: chengen@xmu.edu.cn; Tel.: +86-139-5016-5480

**Abstract:** Diver breathing sounds can be used as a characteristic for the passive detection of divers. This work introduces an approach for detecting the presence of a diver based on diver breathing sounds signals. An underwater channel model for passive diver detection is built to evaluate the impacts of acoustic energy transmission loss and ambient noise interference. The noise components of the observed signals are suppressed by spectral subtraction based on block-based threshold theory and smooth minimal statistic noise tracking theory. Then the envelope spectrum features of the denoised signal are extracted for diver detection. The performance of the proposed detection method is demonstrated through experimental analysis and numerical modeling.

**Keywords:** underwater acoustic signal processing; channel model; signal enhancement; signal denoising; passive detection

## 1. Introduction

A diver is an underwater swimmer who carries a self contained underwater breathing apparatus (SCUBA) system and can stay underwater for a long time. Because of the presence of water, people ashore find it difficult to find, to search for, and to communicate with divers. In addition, when a diver is in danger, the probability of misfortune is high, even with the help of rescuers. There are active and passive sonar system for underwater detection. In shallow water, the active sonar system faces the challenge of reverberation, and the performance requirements of small targets are high. Compared with the active mode, passive sonar has small energy consumption, is cheaper and more hidden, and is being pursued as an alternative [1].

In passive diver detection system, the diver's breathing sound, coming from the gas exchange process in SCUBA, is useful for the passive detection of the diver's presence [2,3]. The periodic pulse characteristic, caused by the vibration of high pressure gas in inhaling [4], is effective to detect the diver's presence. Ref. [5] proposed matched filter to extract periodic characteristic, but reliable reference signal from the diver's breathing sound is hard to obtain. Ref. [6] pre-whiten the noise and detect the diver based on envelope spectrum to a maximum range of 20 m. Although the sounds can be spatially filtered using an underwater array [7], we focus on detecting the presence of diver in a single channel, which also can be used in the multichannel scene.

The performance of passive detection is affected by the underwater environment, mainly including ambient noise interference and transmission loss. The noise spectrum in the ocean is colored by turbulence, rainfall, marine animals, and ships [8]. Since the diver-oriented sound spectrum distributes from hundreds of Hz to more than 75 kHz [7]. Diver detection is mainly affected by wind wave noise from the sea surface [9]. Another difficulty comes from the transmission loss, whose attenuation factors

mainly include water absorption [5], geometric diffusion loss, bottom and surface scattering. In order to predict the characteristics of sound transmission, an acoustic rays model is mostly adopted [10].

Due to low signal to noise ratio (SNR) of observed signals, noise suppression is necessary for detection system, includes noise spectral estimation and noise removing steps. There are many ways used to estimate noise spectral power. Minimum statistics algorithm tracks the minima values of a smoothed power estimate of the noisy signal [11]. Cohen further combined the minimum tracking and the recursive averaging, proposed minima-controlled recursive averaging algorithm (MCRA) [12] and improved algorithm (IMCRA) [13]. Hendriks proposed the subspace noise tracking algorithm (SNT) [14] to search for the signal dimension number and to estimate the noise spectral power in each subspace. Then, the IMCRA method is adopted because of good performance under low SNR conditions [15]. To remove noise from noisy signals, the block-based threshold algorithm (BT) [16] is adopted. Compared with others noise suppression methods, such as random matched filtering [17], cepstral minimum mean-square error motivated noise suppress [18], wavelet threshold [19], the BT method can adaptively estimate the best noise reduction coefficient on time-frequency point at low SNR [20]. The BT method minimizes Stein's unbiased risk estimator (SURE) [21,22] to obtain adaptive block area size and threshold level. It means that the estimated attenuation coefficients of center point in blocks are the results of operation of others points in the blocks.

The present work will focus on diver passive detection, and underwater acoustic channel model from sound source to hydrophone. Firstly, the model of transmission loss and ambient noise is built to evaluate the measured SNR of observed diver's breathing sounds. Secondly, we introduce an adaptive noise subtraction approach to enhance the diver's breathing sounds, which does not need prior knowledge of signals. The ambient noise is suppressed by spectral subtraction approach which is based on BT theory and IMCRA method. Then, extract the envelope spectrum of diver breathing signal for basis feature of diver detection. Finally, detection performance is proved by practical experiment and numeral analysis.

The rest of the paper is organized as follows. Section 2 introduces the acoustic channel model about transmission loss and ambient noise. Section 3 presents detection approach algorithm including noise estimation algorithm, BT algorithm for noise subtraction, envelope spectrum detection method. In Section 4, data acquisition experiment and source signal analysis are introduced. Then, Section 5 evaluates the SNR of measurement of diver signals through underwater channel and the performance of the noise subtraction for detection. Finally, the conclusions are given in Section 6.

## 2. Underwater Acoustic Channel Model

In underwater acoustic environments, the relationship between received sound level (RL) and source sound level (SL) follows passive sonar equation $RL = SL - TL + NL$. SL represents the diver breathing sound level, is related to measuring in standard range (1 m). TL is transmission loss and NL is ambient noise level at hydrophone. As Figure 1 shows, transmission loss and ambient noise are the main parts of underwater acoustic channel model for diver detection.

The acoustic energy transmission loss of the diver breathing soundwave is divided into three kinds as geometric diffusion loss, water absorption loss and scattering loss. In order to predict the transmission loss, the normal mode model and the ray model are often used to model the acoustic transmission process. Considering that the ray model is more suitable for simulating the scene of high frequency signal detection in short distance, we use it to model the underwater transmission of diver breathing sounds. The received signal $R(t)$ can be expressed as

$$R(t) = \sum_{i=1}^{L} \alpha_i A_i \delta(t - \tau_i) \tag{1}$$

where $L$ is the number of intrinsic rays, $A_i$ is the amplitude of $i$th ray and $\alpha_i$ represents attenuation coefficient. $\tau_i$ is the time delay of each ray. Diver breathing sound is regarded as a point sound

source, and the sound wave diffuses in the form of spherical wave, that is, geometric diffusion loss. Water absorption loss is related to the temperature, salinity, PH, frequency, the distance of hydrophone. An experience formula Thorp [5] of predicting the absorption coefficient can be expressed as

$$\alpha(f) = \frac{0.1f^2}{1+f^2} + \frac{40f^2}{4100+f^2} + 2.75 \times 10^{-4}f^2 + 0.003 \tag{2}$$

where $f$ is signal frequency in kHz. Scattering attenuation is due to the scattering of sound waves by the uneven and rough surface of the sea bottom and the sea surface, which leads to the attenuation of sound waves.

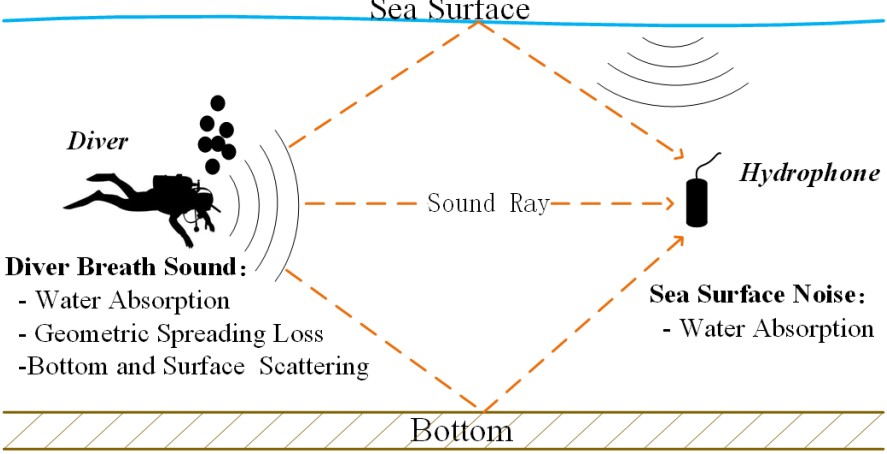

**Figure 1.** Underwater acoustic channel model for diver detection. Transmission loss contains geometry diffusion loss, water absorption and scattering by bottom and surface. Observed signals are affected by ambient noise, for example, wind noise from sea surface.

Besides, ambient noise is also essential in underwater acoustic channel model. Wind noise and ship noise are the main noise in ambient noise. The frequency of the diver's breathing sound we are concerned about is more than 2 kHz. While the ship noise spectrum power is mainly distributing below 200 Hz [23], the ship noise can be ignored. The ambient noise is mainly wind noise above 1 kHz [24]. The wind noise is caused by the vibration of bubbles when the waves hit the sea surface. The designed noise generator uses logarithmic relationship between wind speed and ambient noise level, which is given as [25]

$$\log N_w(f) = 5 + 0.75w^{1/2} + 2\log f - 4\log(f + 0.4) \tag{3}$$

where $f$ denotes sound frequency in Hz, $w$ is wind speed in m/s, $N_w$ is ambient noise level in dB. In the process of transmission, wind noise is also affected by water absorption attenuation. If the scattering of sound waves from the bottom of water is ignored, the transmission loss of wind noise is expressed as [26]

$$TL_{noise} = \alpha_w \times d \tag{4}$$

where $TL_{noise}$ denotes the transmission loss of wind noise in dB, $\alpha_w$ is the attenuation coefficient in dB/km, $d$ is the hydrophone depth in km.

## 3. Noise Reduction and Detection Methodology

This section describes the diver detection process, including noise suppression theory and envelope spectrum detection theory. The framework of proposed diver detection method demonstrates in Figure 2.

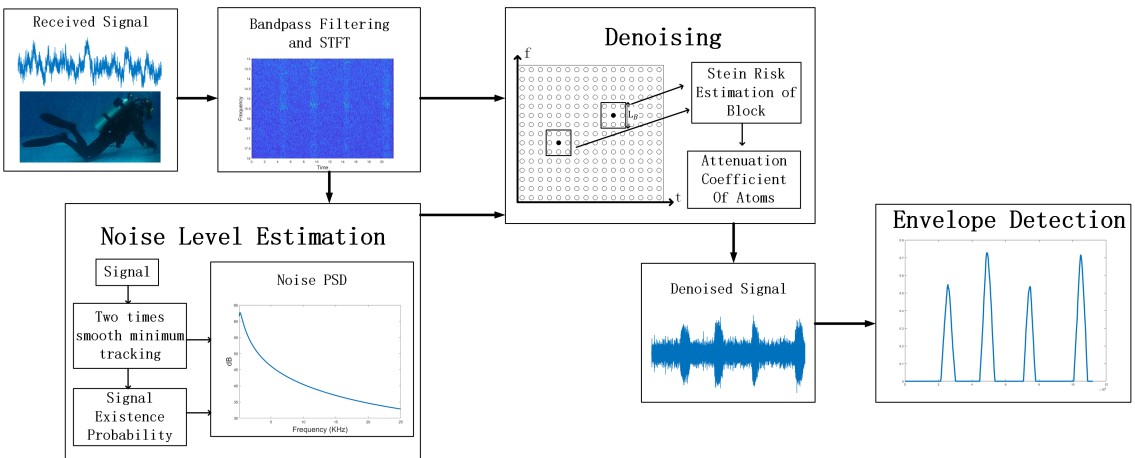

**Figure 2.** Framework of diver detection method.

### 3.1. Noise Reduction

Set $y$ as observed time series of noisy signals. By short time Fourier transform (STFT), time series are decomposed into a family of time-frequency atoms $Y(k,l)$, where $k$ and $l$ are time and frequency scale. In time-frequency domain, the principle of spectral subtraction is to shrink time-frequency points by attenuation coefficient $\alpha_{kl}$. The purpose of $\alpha$ value design is to remove the noise components and keep the signal components. Then, the enhanced signal in time-frequency domain $\widetilde{Y}_{kl}$ is given as

$$\widetilde{Y}_{kl} = \alpha_{kl} Y_{kl} \tag{5}$$

To obtain effective $\alpha_{kl}$, surrounding points of $Y(k,l)$ are divided into a block area. Then, the $\alpha_{kl}$ is given as

$$\alpha_{kl} = (1 - \frac{\lambda}{\gamma_{B_{kl}}})_+ \tag{6}$$

where $\lambda > 0$ denotes the threshold that decides signals presence or not, operation $(g)_+ = max(g,0)$, $B_{kl}$ is block area at point $(k,l)$. Assuming noise power is known and is $\delta^2$, $\gamma$ is the posterior SNR which is given as $\gamma_{kl} = Y^2(k,l)/\delta^2$. Equation (6) demonstrates that the denoising performance of the $\alpha$ is related to block size $L_B$ and threshold level $\lambda$. Because pure reference signal $Y_{pure}$ is unknown, the Stein unbiased risk estimation (SURE) [21] algorithm is used to estimate risk equation given as [16]

$$\widetilde{R}_i = \sum_{l,k \in B_i} E|Y_{pure}[k,l] - a_i Y[k,l]|^2$$

$$\underset{=}{\overset{SURE}{=}} L_B^2 + \sum_{n=1}^{L_B^2} ||h_n(\gamma_n)||^2 + 2 \sum_{n=1}^{L_B^2} \frac{\partial h_n(\gamma_n)}{\partial \gamma_n} \tag{7}$$

where $\gamma_n$ denotes $n$th point in block $B_i$. Function $h_n(\gamma_n)$ is given as

$$h_n(Y_n) = S_n - Y_n = \begin{cases} -\frac{\lambda^2}{S_n^2} \cdot \gamma_n & (S_n > \lambda) \\ -\gamma_n & (S_n \leq \lambda) \end{cases} \tag{8}$$

where $S_n = \alpha_n Y_n$. Then, the square equation and the derivative equation of $h_n$ are given as

$$|h_n(Y_n)|_2^2 = \begin{cases} \frac{\lambda^4}{S_n^4}(\frac{Y_n}{\sigma_n}))^2 & (S_n > \lambda) \\ (\frac{Y_n}{\sigma_n})^2 & (S_n \leq \lambda) \end{cases} \tag{9}$$

$$\frac{\partial h_n(\gamma_n)}{\partial \gamma_n} = \begin{cases} -\lambda^2 \frac{S_n^2 - 2\gamma_n^2}{S_n^4} \cdot \gamma_n^2 & (S_n > \lambda) \\ -1 & (S_n \leq \lambda) \end{cases} \tag{10}$$

In Equation (7), the SURE risk is close to the minimum value in the iterative of $B_i$. The block size $L_B$ must be close in the way that the signal and the noise have slow variations inside the blocks. If the noise is color, e.g., ocean ambient noise, the risk estimator can be near unbiased with a narrow frequency band block [16].

### 3.2. Noise Level Estimation

The discussion in the previous section assumed the noise level to be known. However, the prior information of ambient noise can not be known. We use the IMCRA approach [13] to get the posterior estimation of noise level. In time-frequency domain, the noise power $\sigma^2$ is estimated from statistical average of the noise spectrum power of the past time scale, which is given as

$$\widetilde{\sigma}_d^2(k, l+1) = \widetilde{\alpha}_d(k,l)\widetilde{\sigma}_d^2(k,l) + (1 - \widetilde{\alpha}_d(k,l))|Y(k,l)|^2 \tag{11}$$

where $\hat{\alpha}_d(k,l)$ denotes time-varying and frequency independent smooth parameter, which is given as

$$\widetilde{\alpha}_d(k,l) = \alpha_d + (1 - \alpha_d)p(k,l) \tag{12}$$

where $\alpha_d$ denotes scalar smoothing parameter, $p(k,l)$ is the presence probability of useful signals, which is given as

$$p(k,l) = (1 + \frac{q(k,l)}{1-q(k,l)}(1 + \xi(k,l))\exp(-v(k,l)))^{-1} \tag{13}$$

where $q(k,l)$ denotes signal absence probability, $v(k,l) = frac\gamma\xi1 + \xi$, $\gamma$ and $\xi$ are the posterior SNR and priori SNR, which are given as

$$\gamma(k,l) = \frac{|Y(k,l)|^2}{\sigma_d^2(k,l)} \tag{14}$$

$$\xi(k,l) = \alpha G_{H_1}^2(k,l-1)\gamma(k,l-1) + (1-\alpha)\max\{\gamma(k,l),0\} \tag{15}$$

where $\alpha$ denotes a weighting factor controlling the balance between noise reduction and signal distortion, $G_{H_1}$ is spectral gain function. To estimate $p(k,l)$ robust, signal absence probability $q(k,l)$ is estimated by two iterations of smoothing and minimum tracking. The smoothing in iterations takes into account the strong correlation of neighboring frames in independent frequency bins by a first-order recursive averaging. In first iteration, frequency smoothing of each frame is defined by

$$S(k,l) = \alpha_s S(k,l-1) + (1-\alpha_s)S_f(k,l) \tag{16}$$

where $\alpha_s(0 < \alpha_s < 1)$ denotes smoothing parameter for adjacent frame, $S_f(k,l)$ is the spectrum power of the noisy signal given as

$$S_f(k,l) = \sum_{i=-w}^{w} b(i)|Y(k-i,l)|^2 \tag{17}$$

where $b$ is a normalized window function of length $2w + 1$, e.g., Hanmming window. Then, track the local minimal frequency bins in consecutive time frame with a window size $D$, which is given as

$$S_{min}(k,l) = minS(k,l')|l - D + 1 <= l' <= l \tag{18}$$

In the first iteration, a rough estimation of signal presence $I(k,l)$ is defined as

$$I(k,l) = \begin{cases} 1, \ if \ \gamma_{min}(k,l) < \gamma_0 \ and \ \zeta(k, \ l) < \zeta_0, (signal \ is \ absent) \\ 0, \qquad\qquad otherwise(signal \ is \ present) \end{cases} \tag{19}$$

where $\gamma_0$ and $\zeta_0$ is threshold that use $\gamma_0 = 4.6$ and $\zeta_0 = 1.67$ typically. $\gamma_{min}$ and $\zeta$ denote posterior SNR and priori SNR in minima tracking of first iteration, which are given as

$$\gamma_{min}(k, \ell) = \frac{|Y(k, \ell)|^2}{B_{min}S_{min}(k, \ell)}; \quad \zeta(k, \ell) = \frac{S(k, \ell)}{B_{min}S_{min}(k, \ell)}. \tag{20}$$

where $B_{min}$ is the bias of minimum estimation. Then, in the second iteration, the smoothing process is similar with the first iteration. The spectrum power of the noisy signal is installed as

$$\tilde{S}_f(k,l) = \begin{cases} \frac{\sum_{i=-w}^{w} b(i)I(k-i,l)|Y(k-i,l)|^2}{\sum_{-w}^{w} b(i)I(k-i,l)} \\ \tilde{S}(k,l-1), otherwise \end{cases} \tag{21}$$

The signal absence probability $\tilde{q}(k,l)$ is equation of updated $\gamma_{min}$ and $\zeta$, as

$$\hat{q}(k,l) = \begin{cases} 1 \qquad, \ if \ \tilde{\gamma}_{min}(k,l) \leq 1 \ and \ \tilde{\zeta}(k,l) < \zeta_0 \\ \frac{(\gamma_1 - \tilde{\gamma}_{min}(k,l))}{(\gamma_1 - 1)}, \ if \ 1 < \tilde{\gamma}_{min}(k,l) < \gamma_1 \ and \ \tilde{\zeta}(k,l) < \zeta_0 \\ 0 \qquad, \qquad\qquad otherwise \end{cases} \tag{22}$$

where $\tilde{\gamma}_{min}$ and $\tilde{\zeta}$ denote posterior SNR and priori SNR in minima tracking of second iteration. $\gamma_1$ is threshold that use $\gamma_1 = 3$ typically. In Equation (22), the threshold processing of $\tilde{\gamma}_{min}$ and $\tilde{\zeta}$ guarantees the performance of ambient noise estimation in the presence of weak signals.

### 3.3. Detection Method

Previous research has shown that frequency sub-band envelope spectrum detection (ESD) is an effective detection method to detect the presence of diver [3,6]. ESD takes $D_{env}$ as the feature of the diver's breathing sound, where $D_{env}$ denotes envelope spectrum energy in the range of typical human breathing rates 0.3 Hz–1 Hz. $D_{env}$ takes large value when diver is present, otherwise takes small value. Because ambient noise not affect the envelope spectrum in the range of 0.3 Hz–1 Hz, $D_{env}$ is useful even in the severe ambient noise [3].

Figure 3 shows the calculation process of $D_{env}$. We first extract the envelope of noise-reduced signal. The envelope has obvious periodic characteristic if diver can be detected, otherwise the envelope is random and irregular. Secondly, we transform the envelope into a spectrum. The periodic characteristic of the envelope has a related peak in the spectrum. Since human breathing rates vary with the human body state, e.g., fast swimming or slow swimming, the peak can appear in each position of typical human breathing rates 0.3 Hz–1 Hz. Then, integrate spectrum over 0.3 Hz–1 Hz range to calculate $D_{env}$ for detection.

The results of detection are represented by detection probability $P_D$, which is given as

$$P_D = \begin{cases} 1, & if & D_{env} > 2T \\ \frac{D_{env}-T}{D_{env}}, & if & T < D_{env} <= 2T \\ 0, & if & D_{env} <= T \end{cases} \tag{23}$$

where $T$ denotes threshold of diver detection. The selection of detection threshold is related to the level of ambient noise. We use the $T = D_{env}^N + \varepsilon$, where $D_{env}^N$ is calculated by the noise signal, $\varepsilon$ denotes a positive constant.

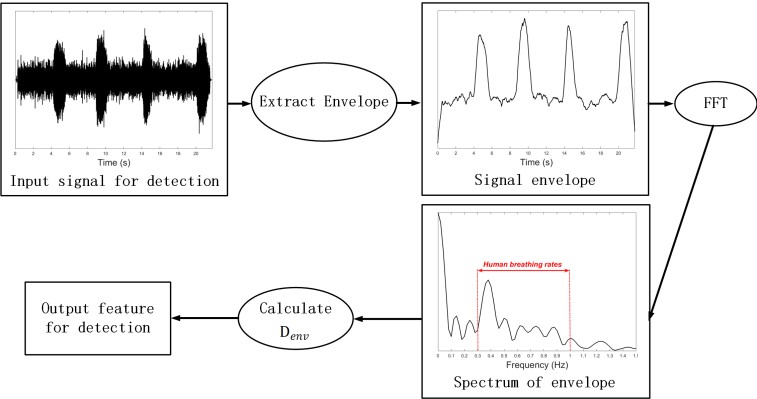

**Figure 3.** Flow chart of calculating $D_{env}$ from signals.

---

**Algorithm 1** Diver detection algorithm BIED based on BT and IMCRA.

---

**Input**: Observed signal

STEP 1: Bandpass filtered signal and STFT. Separate signal into many time frames.

STEP 2: For time frame $l$, compute posterior SNR $\gamma(k, l)$ as Equation (14) and prior SNR $\widetilde{\xi}(k, l)$ as Equation (15)

STEP 3: Compute the first iteration of smoothing power spectrum $S(k, l)$ as Equations (16) and (17), track the minimum $S_{min}(k, l)$ as Equation (18).

STEP 4: Compute minima tracking noise's posterior SNR $\gamma_{min}$ and priori SNR $\xi$ as Equation (20).

STEP 5: Compute a roughly decision about signal presence $I(k, l)$ as Equation (19).

STEP 6: Install noise power spectrum $\widetilde{S}_f(k, l)$ as Equation (21).

STEP 7: Repeat the STEP 3–4.

STEP 8: Compute signal absence probability $\widetilde{q}(k, l)$ as Equation (22). Compute signal presence probability $p(k, l)$ as Equation (13).

STEP 9: Compute smooth parameters $\widetilde{\alpha_d}(k, l)$ as Equation (12).

STEP 10: Estimate noise power $\sigma^2$ as Equation (11).

STEP 11: Compute $h_n(\gamma n)$, $|h_n(\gamma_n)|_2^2$, $\partial h_n / \partial(\frac{\gamma_n}{\sigma_n})$ as Equations (8)–(10).

STEP 12: Compute risk in $i$th block as Equation (7), estimate threshold $\lambda$ and block size $L_B$ by iteration in blocks.

STEP 13: Compute attenuation coefficient $\alpha_{k,l}$ of atoms in time-frequency plane as Equation (6), obtain denoising signal $\widetilde{Y}_{kl}$ as Equation (5).

STEP 14: Transform the time-frequency representation into time series by inverse STFT.

STEP 15: Extract the envelope form result signals. Calculate $D_{env}$ on envelope spectrum from 0.3 Hz to 1 Hz.

STEP 16: Calculate detection probability using Equation (23).

**Output**: Probability of the diver's presence.

---

In summary, the proposed diver detection method reduces noise based on BT and IMCRA, detecting the diver by feature from an envelope spectrum. We call it the BIED method. The detailed steps of the detection algorithm is shown in Algorithm 1.

## 4. Data and Analysis

The data of diver breathing sounds is collected in the swimming pool. The diver assisting in the experiment has more than five years of diving experience. In the experiment, a data acquisition card and a hydrophone were used to record underwater sounds. Figure 4 shows the diver equipped with

SCUBA system breaths underwater. The hydrophone is about 1m away from the diver. The sample rate is 50 kHz.

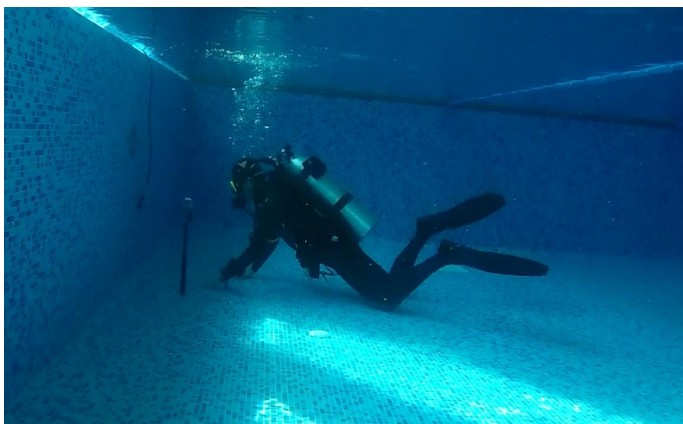

**Figure 4.** In experiment, one channel data acquisition system is used to record the diver's breathing sound underwater. Sample rate is 50 kHz.

Diver breathing sounds come from the air flow in the SCUBA system. The air flow process is controlled by the diver breath. The time series of the diver's breathing sound clearly shows the whole breathing process as Figure 5a shows. Through 2 kHz high pass filter and low pass filter, the inhaling and exhaling sounds can be separated as Figure 5b,c show. In Figure 6, the inhaling sounds frequency distribute in the range of 2 kHz–25 kHz. The frequency of exhaling sounds is mainly below 2 kHz. The inhaling sound and the exhaling sound can represent the diver's breathing process separately. Since the inhaling sounds have better pulse characteristic, while the waveform of exhaling sound is irregular. We use inhaling sound as the interested signal to diver detection.

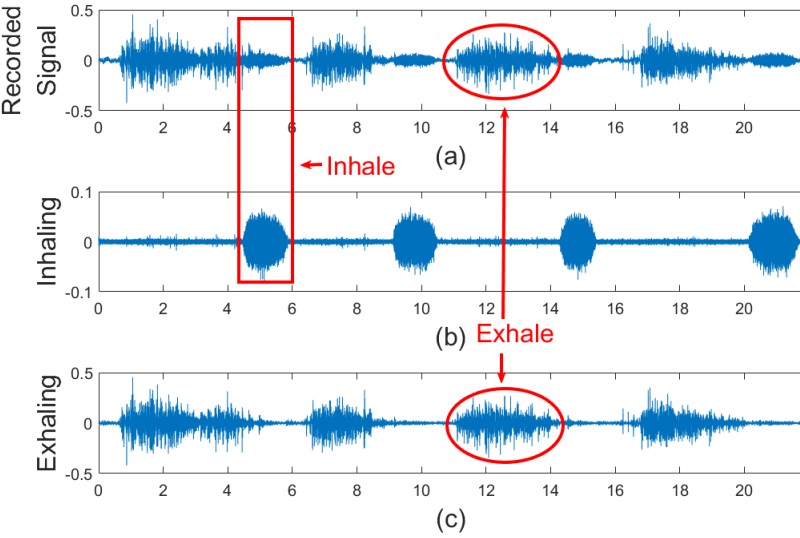

**Figure 5.** Breathing Sound recorded in experiment. The inhaling and exhaling sound are separated by high-pass and low-pass filters with 2 kHz cutoff frequency. (**a**) original recorded signal; (**b**) high frequency inhaling part of signal; (**c**) low frequency exhaling part of signal.

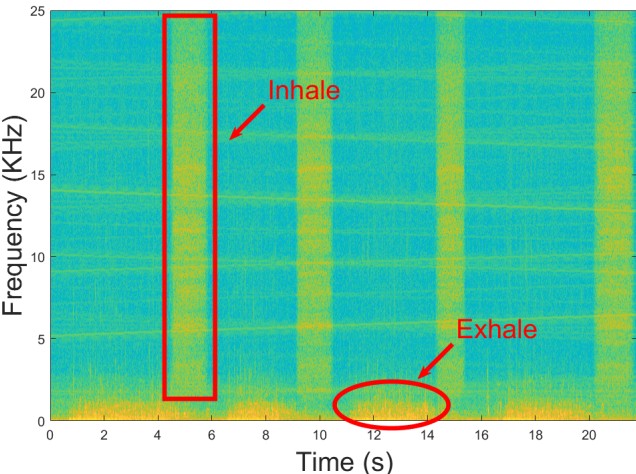

**Figure 6.** The spectrum of the diver's breathing sound. Inhaling sound frequency distributes in 2 kHz–25 kHz when sample rate is 50 kHz and exhaling sound frequency power is below 2 kHz.

## 5. Results and Analysis

### 5.1. Impacts of Underwater Environment

The main impacts of the underwater environment on diver detection are transmission loss and ambient noise interference. The above impacts are taken into account in the established underwater acoustic channel model for diver detection. Then, we can observe the change of breathing sound with channel parameters. Because the diver breathing sounds collected in the experiment have very obvious human breath rate characteristics, we regard them as source signals. Transmission loss is considered to be the result of geometry diffusion loss and water absorption loss. Because scattering attenuation has little effect on signal strength in short distance, we ignored scattering loss caused by bottom and surface. The diver detection environment is set as follows, source depth and receiver depth are 5 m, seafloor depth is 100 m, ambient noise related wind speed is 5 m/s. The Bellhop tool [27] is applied to calculate the attenuation coefficient of independent frequency. In the operations of Bellhop, the sound is modeled as Gaussian rays and is tracked by the sound rays at different incident angles from $-80°$ to $80°$. The ambient noise is considered to be slowly changing, and the associated sea surface wind speed is 5 m/s.

In Figure 7, the power spectral density (PSD) of source sound and attenuated sounds at the distance of 10 m, 30 m, 100 m are shown. With the increase of distance, the sound intensity of diver breathing sound decreases fast. At a distance of 100 m, the attenuation coefficient is close to 35 dB. Compared with the source signal, the acoustic signal attenuates nearly 20 dB at the distance of 10 m, nearly 30 dB at the distance of 30 m. That means the trend of sound intensity attenuation decreases exponentially. Therefore, transmission loss is mainly due to geometry diffusion loss in 100 m, and frequency dependent water absorption loss has little effect on signal attenuation. The frequency is not a major limitation in selecting sub-band for diver detection in 100 m.

Figure 8 shows the ambient noise, source sound and observed signals at the distance of 10 m, 30 m, and 100 m. Because of the effect of strong noise and strong attenuation, the observed signals have lost the waveform of source sound even at the distance of 10 m. Therefore, the first task of detection is to find the significant sub-band of the signal. The observed signals are divided into several sub-bands to discuss the effects of attenuation and noise, including 3 kHz–8 kHz, 8 kHz–13 kHz, 13 kHz–18 kHz and 18 kHz–23 kHz. Figure 9 compares the SNR of each sub-band. The SNR of sub-band 3 kHz–8 kHz is the lowest because the PSD of ambient noise is high in this frequency band. Otherwise, the SNR of other sub-bands are similar. We choose sub-band 13 kHz–18 kHz for diver detection because of the higher SNR.

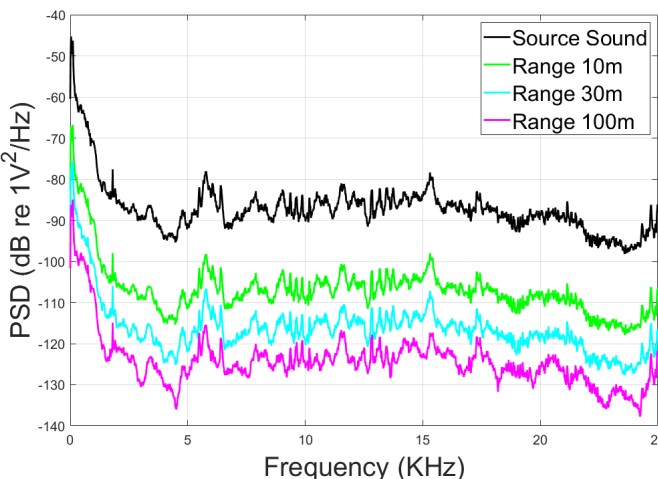

**Figure 7.** PSD of source sound and observed signals at the range of 10 m, 30 m, 100 m.

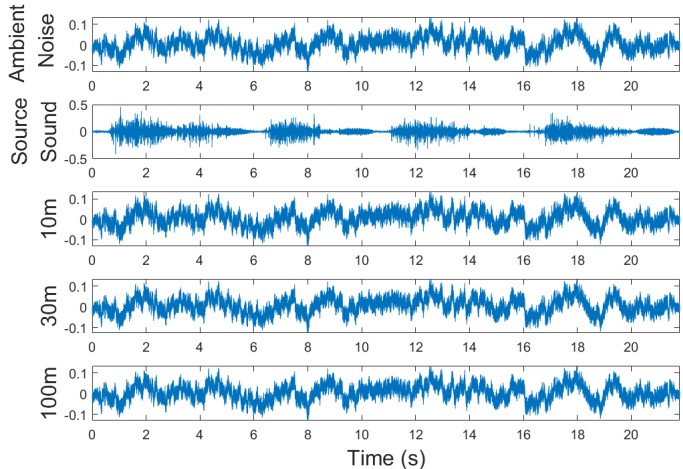

**Figure 8.** Ambient noise, source sound and observed signals at the range of 10 m, 30 m, 50 m.

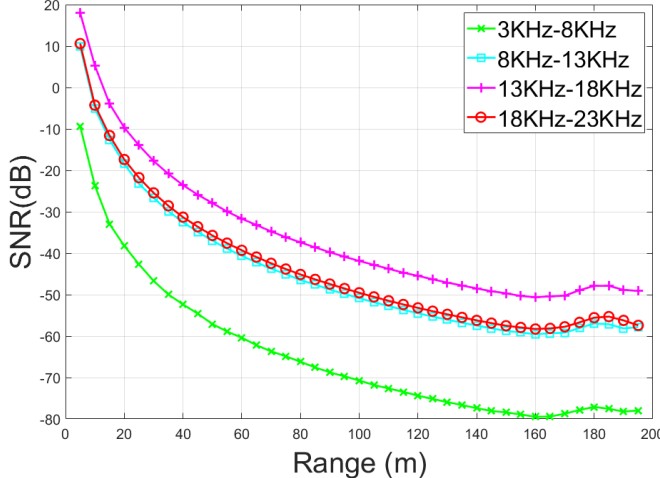

**Figure 9.** SNR of frequency band 3 kHz–8 kHz, 8 kHz–13 kHz, 13 kHz–18 kHz and 18 kHz–23 kHz. The 13 kHz–18 kHz band has the best SNR performance.

*5.2. Detection System Performance*

The detection of the underwater diver is affected by the underwater environment. For example, in a river or harbor, the environmental noise will cause the received SNR to decrease. We verify the performance of the detection system by adjusting the SNR. It is assumed that the ambient noise level is controlled by the wind and waves noise with 5 m/s wind speed, and the SNR can be changed by changing the detection distance. The proposed BIED method firstly uses SME theory and BT theory to estimate the ambient noise level and to remove the noise. Then, extract the characteristic value $D_{env}$ from the envelope spectrum to detect the presence of a diver. The threshold of diver detection is set to $T = D_{env}^N + D_{env}^N/3$.

To evaluate the SNR of the denoised signal, an evaluation value $SNR_M$ is defined as

$$SNR_M = 10 \log \frac{\sum y(n) \times M(n)}{\sum y(n) \times |M(n) - 1|} \tag{24}$$

where $M$ denotes the manually marked presence position of diver breathing sounds, $|M - 1|$ is the opposite of $M$. In sequence $M$, the signal presence position is marked as 1, otherwise 0. The $SNR_M$ represents the ratio of diver breathing sound presence signal component and absence signal component in time series. High SNR means that the envelope characteristics of diver breathing sound are more obvious and the $D_{env}$ is high.

The length of time series also affects $D_{env}$. Theoretically, the larger the number of diver's breathing cycles contained in the observation window, the larger the corresponding detection value $D_{env}$. However, the long observation window does not meet the real detection requirement with reliability and timeliness. For example, when a diver is escaping from the hydrophone, a short window must be used to capture the presence of the diver in time. Hence, we use a time window of 22 s to detect diver, which contains four breathing periodic pulse at least.

Figure 10 compares the pre-processed signals of ESD method and the ones of proposed BIED method at the distance of 10 m and 30 m. The pre-processed signal of BIED has stronger inhaling sound pulse than the ESD's in high SNR condition as Figure 10a,b show. At the distance of 30 m, Figure 10d shows that the enhanced signal in BIED has inhaling sound characteristics, while the observed signal in ESD is almost submerged by noise as Figure 10c shows.

In Figure 11, the $SNR_M$ of pre-processed signals in the ESD method and the proposed BIED method are compared. The curve of BIED method has higher $SNR_M$ value than the curve of ESD method within a distance of less than 55 m. That proves the noise elimination process in BIED is effective to enhance the observed diver breathing sound. In the low SNR conditions, the noise elimination method is difficult to distinguish the background noise component form the observed signals. Then, two methods have approximate $SNR_M$ value at a long distance.

In Figure 12, two curves show that the detection probability decreases as detection distance increases. The proposed BIED method has a higher detection probability in the near range. The reason for this is that the noise reduction process further enhances the SNR of 13 kHz–18 kHz band signal. The ESD method detects the diver to a maximum range of near 20 m, which is similar to the detection results of Johansson [6]. Compared with that, the BIED method can detect diver until the 40 m range.



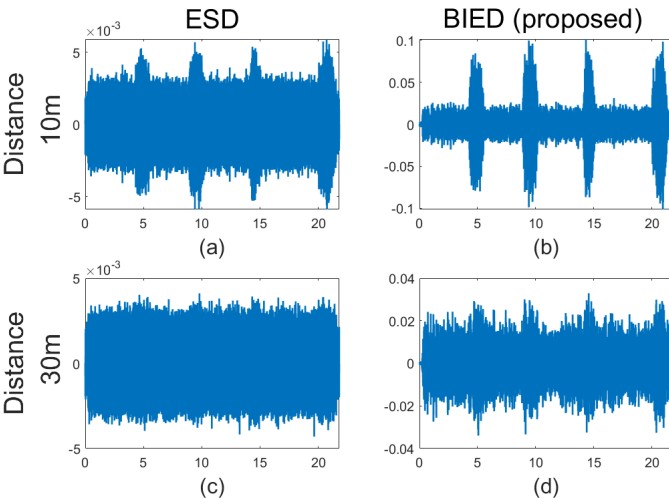

**Figure 10.** Pre-processed signals in the ESD method and the BIED method. (**a**) ESD at the distance of 10 m; (**b**) BIED at the distance of 10 m; (**c**) ESD at the distance of 30 m; (**d**) BIED at the distance of 30 m.

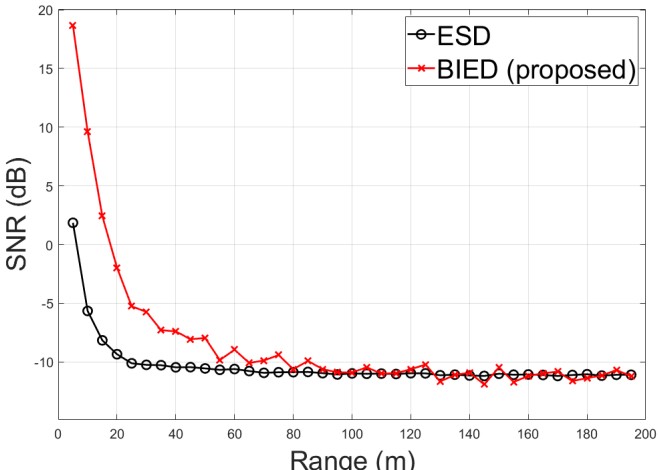

**Figure 11.** SNR of pre-processed signals in the ESD method and the BIED method.

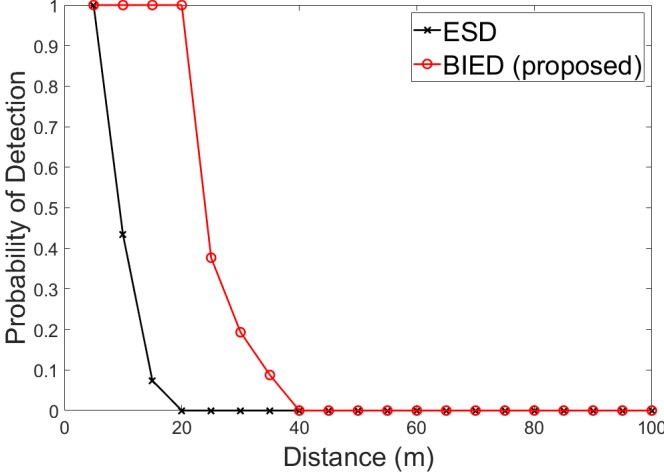

**Figure 12.** Detection probability. The detection threshold is set to $T = D_{env}^N + D_{env}^N/3$.

## 6. Conclusions

In this paper, we propose a diver detection method BIED based on suppressing ambient noise and extracting envelope spectrum features. The built acoustic channel model mainly considers transmission loss and noise interference in the underwater passive detection scenario. In the numeral analysis, the 13 kHz–18 kHz band of observed signals is selected for diver detection. While the ESD method can detect a range up to 20 m, the proposed BIED method detects one diver to a maximum range near 40 m.

Although our work shows effectiveness in diver detection, there are still many challenges to face. One of them is that the strength of the target sound source is too weak and easily covered by noise, which is the mainly reason for limiting detection distance. There is also a need to detect multiple divers' presences. We are working to achieve passive detection in these challenges.

**Author Contributions:** Conceptualization, Q.T. and F.Y.; methodology, Q.T.; software, Q.T.; validation, Q.T.; formal analysis, Q.T.; investigation, Q.T.; resources, F.Y. and Q.T.; data curation, W.Y. and F.Y. and Q.T.; writing–original draft preparation, Q.T.; writing–review and editing, Q.T.; visualization, Q.T.; supervision, F.Y.; project administration, E.C.; funding acquisition, E.C. All authors have read and agreed to the published version of the manuscript.

**Funding:** This research was funded by National Natural Science Foundation of China (61571377, 61771412, 61871336) and the Foundamental Research Funds for the Central Universities (20720180068).

**Conflicts of Interest:** The authors declare no conflict of interest.

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
