# Peer review of "An Approach for Diver Passive Detection Based on the Established Model of Breathing Sound Emission"

_jmse, doi:10.3390/jmse8010044_

Round 1

Reviewer 1 Report

The paper introduces an approach for detecting the presence of divers based on their acoustic signature. The outcome of the experiment suggests that this method performs well at detecting the presence of divers at further distances than some existing techniques.

More discussion is needed on other factors that affect the performance of the methods (not just the distance between the diver and observer). For instance, would this method work if there are multiple divers in the vicinity, what is the length of the audio sample (in seconds) needed to effectively identify a breathing pattern, the experiment was conducted in a swimming pool which is likely to have different ambient noise profile to a river/harbour?

In general, the document is well-organised, however, the writing (grammatical issues, poor phrasing) throughout the paper needs some improvements.

A sentence or a paragraph (maybe in the Introduction or Section 2) outlining the need for/the benefits of a passive diver detection (e.g. for threat detection) versus active sensing technologies is missing.

Other minor comments:

On line 140, “Figure 9 shows the ambient noise, source sound and observed signals at the distance of 10m, 30m, 141 100m”

Should this be Figure 8?

Author Response

Authors Responses to Reviewer 1

We wish to thank the reviewer for the comments and suggestions that helped us revise the manuscript for re-submission. According to your comments, our major revision of the manuscript is as follows:

Reviewer1:

The paper introduces an approach for detecting the presence of divers based on their acoustic signature. The outcome of the experiment suggests that this method performs well at detecting the presence of divers at further distances than some existing techniques.

 The paper introduces an approach for detecting the presence of divers based on their acoustic signature. The outcome of the experiment suggests that this method performs well at detecting the presence of divers at further distances than some existing techniques.

 More discussion is needed on other factors that affect the performance of the methods (not just the distance between the diver and observer). For instance, would this method work if there are multiple divers in the vicinity, what is the length of the audio sample (in seconds) needed to effectively identify a breathing pattern, the experiment was conducted in a swimming pool which is likely to have different ambient noise profile to a river/harbour?

 Thank you for your suggestion. The discussion part of the paper is indeed relatively simple. Here are the changes that take your example:

The detection of multiple divers will be our further work. In this paper, we aim at single diver detection. So, we make that clear in Abstract (line 2), in 4.2 Detection System Performance (line 162), in Conclusion (line 196). We also state that in Future Work (line 199).

The discussion of the length of the audio sample is added in 4.2 Detection System Performance (line 169 – line 174). I understand the length of the audio sample as the length of the observation window. We choose 22s lengths of the window. Because it contains four breathing periodic pulse at least. And, a longer observation window reduces the timeliness and the reliability of detection.

The ambient noise in a river and harbor is really different from that in the pool. We add the wind and waves noise with 5m/s wind speed to simulate a general ambient noise profile. The description of ambient noise is added in 4.2 Detection System Performance (line 156 – line 160).

 In general, the document is well-organised, however, the writing (grammatical issues, poor phrasing) throughout the paper needs some improvements.

 Thank you for your suggestion. I have tried best to check the manuscript for grammar, style and syntax.

 A sentence or a paragraph (maybe in the Introduction or Section 2) outlining the need for/the benefits of a passive diver detection (e.g. for threat detection) versus active sensing technologies is missing.

The compare of active and passive sonar is added in Introduction (line 16 – line 20)

 In line 140, “Figure 9 shows the ambient noise, source sound and observed signals at the distance of 10m, 30m, 141 100m”. Should this be Figure 8?

Thanks for reminding. It has been revised.

Reviewer 2 Report

Units are to be written in normal text and not italics. SNR stands for Sound to Noise Ratio, not Sound to Noise Rate. A reference is missing on page 3. There is no concept of Propagation Loss in underwater acoustics.

Author Response

Authors Responses to Reviewer 2

We wish to thank the reviewer for the comments and suggestions that helped us revise the manuscript for re-submission. According to your comments, our major revision of the manuscript is as follows:

Reviewer2

Units are to be written in normal text and not italics.

The unit in the whole paper has been modified.

 SNR stands for Sound to Noise Ratio, not Sound to Noise Rate.

I'm sorry for the mistake, it has been modified.  (line 36) (line 166)

 A reference is missing on page 3.

Thanks for reminding. The reference is added.

 There is no concept of Propagation Loss in underwater acoustics.

I have searched for related literature. Transmission Loss is more commonly used in underwater acoustics. So, I replaced all Propagation Loss, e.g., in Abstract (line 4) and in the caption of Figure 1.
